# Recent Advances in the Use of Probiotics to Improve Meat Quality of Small Ruminants: A Review

**DOI:** 10.3390/microorganisms11071652

**Published:** 2023-06-25

**Authors:** Sudeb Saha, Kohtaro Fukuyama, Marina Debnath, Fu Namai, Keita Nishiyama, Haruki Kitazawa

**Affiliations:** 1Food and Feed Immunology Group, Laboratory of Animal Food Function, Graduate School of Agricultural Science, Tohoku University, Sendai 980-8572, Japan; kotaro.fukuyama.p8@dc.tohoku.ac.jp (K.F.); fu.namai.a3@tohoku.ac.jp (F.N.); keita.nishiyama.a6@tohoku.ac.jp (K.N.); 2Livestock Immunology Unit, International Education and Research Center for Food and Agricultural Immunology (CFAI), Tohoku University, Sendai 980-8572, Japan; 3Department of Dairy Science, Faculty of Veterinary, Animal and Biomedical Sciences, Sylhet Agricultural University, Sylhet 3100, Bangladesh; 4Ministry of Public Administration, Government of the People’s Republic of Bangladesh, Dhaka 1000, Bangladesh; marina26nath@gmail.com

**Keywords:** probiotics, meat quality, small ruminants

## Abstract

Meat from small ruminants is considered a high quality and delicacy product in many countries. Several benefits have been perceived from probiotics as dietary supplements, such as improved carcass weight, color, tenderness, flavor, muscle fiber structure, water-holding capacity, and healthy fatty acid profile of the meat. Thus, the present review focuses on the effect of probiotics on improving the quality of meat from small ruminants. Though many benefits have been associated with the use of probiotics, the findings of all the considered articles are not always consistent, and the mechanisms behind improving meat quality are not appropriately defined. This variability of findings could be due to the use of different probiotic strains, dosage rates, number of days of experiment, nutrition, breed, age, and health status of the animals. Therefore, future research should emphasize specific strains, optimal dose and days of administration, route, and mechanisms for the specific probiotic strains to host. This review provides a comprehensive overview of the use of probiotics for small ruminants and their impact on meat quality.

## 1. Introduction

The growing world population demand for animal proteins has been increasing substantially. A recent study showed that global meat production was 263 million tonnes in 2018 and is expected to rise to 445 million tonnes by 2050 to meet consumer demand [1]. The demand for small ruminant meat, such as mutton (sheep meat) and chevon (goat meat), consumption has increased significantly due to their nutritional values compared with other animal species [2]. As a result, production of sheep and goat meat with better quality, high nutritional value, and better taste have received special attention from researchers [3]. However, farmers have focused more on improving the health status, production, and feed efficiency of farm animals by using growth promoters in the animal feed [4]. Thus, the use of antibiotics for the treatment and as growth promoters for animals has augmented the creation of antibiotic-resistant microbes, which negatively impact human life and the environment. Recently, the scientific world has discussed avoiding the irrational use of antibiotics for treatment and restricting the use of antibiotics as growth-promoting additives to avoid the risks to meat consumers and the environment. Therefore, many investigators focused on antibiotics substitutes, namely feed additives or supplements as probiotics (as single or mixed strains) to the animal feed [5,6]. Probiotics have been shown to have positive effects on the microbial ecosystem [7], nutrient synthesis [4,8], growth performance [9], carcass weight [10,11], muscle production and meat quality [3,12,13,14], prevention of enteric diseases [15], and immunity [16]. A study showed that 2.5 months of dietary supplement of probiotics (*Lacticasebacillus rhamnosus*; *Saccharomyces cerevisiae*) could maintain microbial balance in the gastrointestinal tracts of weanling lambs [6]. Similarly, other studies revealed that probiotics can improve lambs’ nutrient synthesis and growth performance [8,9]. In addition, using probiotic (*Lacticaseibacillus casei* and *Lactiplantibacillus plantarum*) supplements in lambs exhibited improved meat tenderness and flavor [12]. Hernández-García et al. [9] reported that yeast supplementation showed improved carcass traits in lambs. In contrast, Gadekar et al. [17] reported that dietary supplements of *Saccharomyces cerevisiae* did not affect lamb carcass traits. However, Picard and Gagaoua [18] reported that probiotics could regulate muscle fiber properties, which are directly linked to carcass yield and meat quality traits. Yet, the molecular mechanisms of probiotics related to the development of muscle fiber and modification of the carcass traits are not well documented [19]. Thus, exploring the insight knowledge on the interaction between probiotics and muscle development will help to improve carcass yield and meat quality traits of small ruminants (sheep and goat). Therefore, this review article aims to explore the effects of specific probiotics to improve the carcass and meat quality traits of small ruminants. 

## 2. Methodology

The objective of this review was to collect and uphold the critical insight information available on the use of probiotics use in small ruminant production and their effects on performance and meat quality parameters. An electronic search was conducted using some keywords relevant to this study. The keywords used for the search were: probiotics, performance, meat quality, goat, sheep, and small ruminants. The articles published until 2022 were considered in search results and duplicates were removed. Articles were retrieved from four databases: Google Scholar (https://scholar.google.com/, accessed on 30 November 2022), Scopus (https://www.elsevier.com/ja-jp/solutions/scopus, accessed on 5 December 2022), PubMed (https://pubmed.ncbi.nlm.nih.gov/, accessed on 14 December 2022), as well as Science Direct (https://www.sciencedirect.com/, accessed on 22 December 2022). Information from each selected source was compiled while accounting for the probiotic strain, dosage, host, period of use, and the effect of probiotics on performance and meat quality. Finally, general information, details of treatment, and variable outputs from these articles (19 articles) were summarized in registered in the database of review. The literature was selected based on the following criteria mentioned in Figure 1. 

## 3. Probiotics: Definition, Characteristics, and Use of Microorganisms as Probiotics

Many ancient civilizations used microbes to prepare fermented food items. However, research on the use of microorganisms in food and feed products and their effects on health have been studied recently. The idea of probiotics was first evolved in 1908 by Élie Metchnikoff, a Nobel laureate at the Pasteur Institute, who established a link between health and longevity by consuming beneficial microbes from yogurt [20]. The word “probiotics” was first used in 1965 by Lilly and Stillwell [21] to refer to the substances secreted by microorganisms that stimulate the growth of other microorganisms. The “probiotic” word originated from the Greek word meaning “for life” and has various meanings over the years. During the following decade, the probiotic word was applied by Fujii and Cook [22] and indicated synthetic chemicals in mice that conferred protection against infection due to *Staphylococcus aureus*. Subsequently, in 1974, Parker [23] used the term “probiotic” in a broader sense to refer to microorganisms which have interactions with the host (animal or human), i.e., “organisms and substances, which contribute to intestinal microbial balance”. In 1989, the definition of probiotics was used by Fuller where probiotics refer to a live microbial feed supplement, including *Lactobacillus* species, *Bifidobacterium* species, *Streptococcus* species, yeasts, and molds [24]. Numerous studies concerning probiotics have been published since then. The most commonly used definition of “probiotics” was proposed by FAO and WHO [25]. According to these organizations, probiotics are live microorganisms when administered in an adequate amount to confer a health benefit to the host. Currently, the U.S. Food and Drug Administration declared that probiotics are also safe ingredients for animal use [26]. When probiotics are used in the animal industry, they must have common characteristics (Figure 2), such as the ability to colonize and be metabolically active in the host body; promotion of animal health; applicability to industrial use, and safety to the animal and human body [27]. 

The microorganisms that are generally used as probiotics are presented in Table 1. The *Lactobacillus* and *Bifidobacterium* genera are reported as the bacterial groups found in the gastrointestinal tract (GIT) and commonly used in small ruminants’ nutrition [28]. Additionally, other genera, *Enterococcus* and *Pediococcus,* are normally commensal to the GIT and are widely used for livestock feeding. *Bacillus* sp. are Gram-positive spore-forming microorganisms; though some *Bacillus* species produce toxins to the animal body, some species have a probiotic effect on the host. The yeast, *Saccharomyces cerevisiae,* is widely present in nature and is used as probiotics in the livestock industry [29]. Other microorganisms are also used as probiotics in small ruminants’ nutrition as mentioned in Table 1. Probiotics are used in all stages of small ruminant production. However, the target of probiotic use depends on the situational demands, such as growth promotion, animal health promotion, treatment of the disease, or improvement of the product quality (meat, milk etc.).

## 4. Role of Probiotics on Animal Health and Nutrition

Using probiotics has become popular during the last two decades for their health benefits to animals and humans [31,32]. Probiotics have been used in ruminants as therapeutic [33] and dietary supplements [11] to reduce morbidity and mortality and increase production (meat and milk). Applying single or multi-strain microorganisms at the farm level may act complementarily or synergistically. However, the possible mechanisms of probiotics are not well defined. Several factors, such as strain specific mechanisms on the host and specific actions against certain diseases, influence the loose definition. Some proposed mechanisms by which probiotics act against unwanted microorganisms include contributing to colonizing the beneficiary microbes, reinforcing the intestinal barrier by secretion of mucus, producing antimicrobial compounds, modulating the immune system, and maintaining the beneficiary intestinal composition and their activity.

Like other organic nutrients in the intestine, probiotics are partly digested and broken down. Thus, only small portions are viable. Later, probiotics colonize the intestinal layer, competitively exclude the pathogen, and enhance the nutrient synthesis in the GIT. A study by Chen et al. [34] reported that dietary supplements of probiotics foster rumen microbial protein synthesis in lambs. It is also considered that probiotics compete with other non-beneficiary organisms for the endowment of nutrients to the host body [35]. For instance, *Lactiplantibacillus plantarum* breaches the carbohydrate to simpler compounds such as glucose, which provides energy to the animal [36]. Moreover, *Lactobacillus* can improve muscle development and meat quality by regulating the different mechanistic pathways associated with muscle development [3]. In addition, the use of *Aspergillus oryzae* as probiotics ameliorates the animal performance and body weight by producing different enzymes related to improving fiber digestion and nutrient absorption to the animal [37].

Probiotics combat pathogens by producing different inhibitory substances, such as organic acids, hydrogen peroxide, and bacteriocins [33]. Furthermore, many antibiotic metabolites (aciolin, acidophilin, lactobacillin and lactolin) release from the probiotics, which have inhibitory activities against different pathogenic microorganisms (*Salmonella*, *Shigella*, *Staphylococcus*, *Proteus*, *Klebsiella*, *Pseudomonas*, and *E. coli*) [28]. Additionally, probiotic bacteria can regulate immunomodulatory stimulation of the immune system [38,39] and regenerate intestinal mucosa [40]. Probiotics can increase the concentration of immunoglobulin [41] and augment the activity of macrophages and natural killer cells [42]. Probiotics can modulate the anti and pro-inflammatory cytokine production to control inflammation in the host [43]. For instance, *Lactobacillus strains* can reduce inflammation by downregulating the proinflammatory cytokines (T lymphocytes, *IL-8*, *TNF-α*, *IL-6*) as well as upregulating the anti-inflammatory cytokines (*IL-10*, *TGF*-*β, IL-1Ra*) in the intestine of piglets [44]. 

## 5. Effects of Probiotics on Meat Quality

### 5.1. Effects of Probiotic Supplementation on General Eating Quality Traits

Reports on the effects of probiotic supplementation on the meat quality of small ruminants (sheep and goat) are scarce. Instead, most of the studies focused on growth performance and disease control. Table 2 presents the list of recent studies focused on the effects of probiotics on the meat quality of small ruminants (sheep and goat). Meat quality is a term which impels consumer purchasing decisions and eating experiences. In general, the quality of meat includes the attributes of color, tenderness, juiciness, flavor, and water-holding capacity (WHC) [45]. These attributes can be influenced by animal (animal, breed, sex, diet) and environmental factors (climate, slaughter hygiene and procedure, and preparation of final meat products). As a dietary supplement, probiotics also showed positive effects on carcass weight and meat quality [46,47]. Such effects on meat quality includes improvement of product quality and shelf life [3], upgrading the sensory qualities [12,13], and improving color and tenderness [5,12] and healthy fatty acid profiles [48,49]. The beneficial effects of probiotics supplements on carcass weight and meat quality are exhibited in Figure 3.

Meat color and sensory traits are considered important attributes and indicate consumer purchasing decisions. This is because consumers consider color and flavor for the freshness and wholesomeness of the meat [56]. Meat color and sensory traits depend on pre- and postmortem handling and relate to the amount of myoglobin pigment, hemoglobin, oxidation of lipids, and the pH level. Normally, myoglobin is responsible for the color of meat. Any oxidation or reduction of myoglobin can decrease the color intensity of the meat. A study by Nie et al. [5] found that feeding a mixture of probiotics (*B. licheniformis*, *B. subtilis*, and *L. plantarum*) for 60 days to lambs improved the meat color by reducing the oxidation of myoglobin. As probiotics, *Lactobacillus* sp. (*L. casei* and *L. plantarum*) have shown a positive effect on meat color and flavor [12]. Likewise, *Lactobacillus* and yeast (*Sacchaomyces cerevisiae*) supplements also improved the meat color [6]. In general, volatile compounds are responsible for producing odor and flavor in meat; supplementation of probiotics increases the antioxidant activity, which may reduce the lipid oxidation and thereby improve lamb meat flavor [12]. In recent years, researchers explored pH as an important trait for meat color and its effect on the glycolysis and lactic acid formation during the pre- and postmortem slaughtering stages. In general, pH gradually declines from an initial value of approximately 7.2 to an ultimate pH of about 5.6. However, a rapid decline of pH level in carcasses leads to the development of pale, soft, and exudative meat. In this case, myoglobin (a water-soluble protein) is lost along with water that exudates from the meat, which causes a pale, unsavory appearance in meat. Along with pH, the oxidation of muscle fiber relates to the meat’s color and lightness [12]. Oxidation of myoglobin reduces the redness of meat [5]. Considering the above facts, recently, researchers have explored the potential use of probiotics to improve meat color and stability. For example, Tian et al. [57] reported that dietary supplementation of *Limosilactobacillus reuteri* altered muscle fiber characteristics and can contribute to improving the meat color. Furthermore, feeding *L. reuteri* significantly improves the tenderness of meat by increasing water-holding capacity in muscle. The water-holding capacity of meat is responsible for the tenderness and juiciness of the meat [57]. Diets with probiotic supplements assist to keep the high pH in meat, which improves WHC in meat. Lambe et al. [58] revealed that probiotic supplements are positively impacted by intramuscular fat deposition, which is positively associated with marbling, tenderness, and meat flavor. We hypothesize that probiotic supplements induce adipocyte to develop and aid in depositing fat within the skeletal muscle. Liu et al. [12] reported that probiotic supplements increased the density of fibers in muscle, which causes softness and tenderness of meat. Several other studies reported that probiotic supplements improved meat color, flavor, and tenderness; thus, panelists preferred probiotic supplemented meat to meat without probiotic supplements [48,52]. In addition, dietary probiotic supplements might also alter the composition of volatile compounds, which is linked to the quality attributes of meat [5,12]. In addition, probiotics possess antioxidant properties [59] and aid in improving the poly unsaturated fatty acid profile in meat [60]. Some probiotic strains can produce bacteriocins that act as protective agents against lipid oxidation and keep positive organoleptic attributes in meat. It can be concluded that the use of probiotics improves the quality of meat.

### 5.2. Effects of Probiotic Supplementation on Lipid Oxidation

Beyond the benefits of using probiotics in meat attributes, researchers have also found that using probiotics in fermented meat products combat the growth of pathogenic and spoilage microorganisms. For instance, *L. rhamnosus* GG (10^5^ and 10^7^ colony-forming unit (CFU)/g) exhibited an inhibitory effect on the growth of *Enterobacteriaceae* during the fermentation process of meat [61]. In general, lipid oxidation is a major concern for fresh and fermented meat products, which negatively impacts the product’s sensory attributes and subsequently, its acceptability by the purchaser [62]. Free fatty acids are the main precursors for lipid oxidation in fresh and fermented meats [63,64]. Probiotics can act as protective agents against lipid oxidation because of their ability to produce bacteriocins that inhibit lipolytic microbes from forming free fatty acids [65,66]. Özer et al. [67] reported that the use of *L. plantarum* at 10^5^ CFU/kg in fermented sucuk remarkedly reduced the levels of thiobarbituric acid reactive substances (TBARS), which is a marker of lipid peroxidation, compared to control samples. Similar findings were documented by Trabelsi et al. [68] in which lesser TBARS were measured in minced meat that was inoculated with *L. plantarum*. Additionally, adding *L. acidophilus* and *B. lactis* in fermented meat sausage reduced lipid oxidation and contributed to affirmative organoleptic properties [69,70,71].

Collectively, the findings reveal that dietary supplement of probiotics contributed to ameliorating the meat quality of small ruminants (sheep and goat).

## 6. Safety Concerns Relate to Probiotic Use

When microorganisms are used as probiotics, safety is a significant issue. Not only for the animal’s health but also for human health, probiotics should be nontoxic, non-pathogenic, and not related to antibiotic resistance gene transmission. Much research concerning probiotics are published mainly focusing on efficacy, capacities to improve gastrointestinal health, digestion, and growth performance rather than considering their safety and risks in practical usage. In general terms, there is greater number of articles describing the beneficial effects of the use of probiotics (>80%) rather than the negative effects [72]. However, we must consider the differences in experimental factors, such as animal age, dosage, dosing route/method, breed, sanitary status, probiotic use days, or diets, which are deemed generally safe for animal use. In most cases, probiotics are considered safe, but little evidence exists that probiotics are utterly safe and zero risk does not exist [73]. Safety issues related to probiotics use are mostly based on the *Lactobacillus* and *Bifidobacterium* bacteria [74,75]. These two probiotic genera are barely associated with any negative effect on the host. Any array of microorganisms could be used as probiotics, but uncertainty always exists about the safety issue. Thus, more research are needed on the safety issue of probiotics. Thus, probiotic developers or researchers should pay special attention to some issues related to probiotic safety (FAO, 2016) [76]:(i)Safety assessment of one probiotic or specific probiotic cannot be generalized to other probiotics. Each probiotic requires its own specific safety and risk assessment.(ii)The adverse and severe effects of the probiotics could be context specific, depending on the host’s susceptibility and physiological state.(iii)No probiotics can be issued as 100% safe or zero risk in the case of drug use.(iv)Public awareness related to probiotic safety and risk is limited. There is a need for risk-benefit analysis and information to the user or consumer.

Although microorganisms used as probiotics in animal feed are relatively safe, precautions should be considered to protect the animals from unsafe microorganisms associated with some risks that may occur when using microorganisms as probiotics [76,77]. Some are as follows:o Any infection to the animals when fed the probiotic.o Any infection to the consumers of animal products produced by animals fed probiotics.o Transfer of antibiotic resistance from probiotics to other pathogenic microorganisms.o Release of infectious and pernicious compounds to the environment from the animal production systems.o Chance of transfer infection to the animal and animal feed handlers.o Skin and/or eye and/or mucus membrane sensitization in the handler of probiotics.o Any detrimental metabolic or toxic effects in the host due to production of toxins by probiotics.o Hyper-stimulation of the immune system of the animal.

However, to fulfill the list of risks is very difficult for probiotic developers or researchers in practical aspects. Thus, some of them may consider a priority basis, such as the transfer of antibiotic resistance genes/determinants from some probiotic bacteria, the chance of infection, and the presence or production of toxins by probiotic bacteria [77]. Safety and risk assessments of a particular microorganism are considered when used as probiotics for animal or human use.

## 7. Conclusions

In this review, we summarized the previous studies to provide an overview of the effects of probiotics to improve the meat quality of small ruminants. Probiotics appear as promising feed additives. They are of natural origin and are generally regarded as safe for animals. Although probiotics are safe, it is better to consider further safety issues when used in animal trials and animal products. Using probiotics in the diet of small ruminants may improve the different attributes of meat and muscle fiber properties through different mechanisms in animals and carcasses. Moreover, many studies showed probiotics have positive impacts on improving meat quality, but in some cases, the effect was unclear. Thus, further studies characterizing specific strains, optimal dose, safety concerns, and understanding of the mechanisms to improve the particular traits of meat could help to use more effective probiotics to improve the quality of meat from small ruminants.

## Figures and Tables

**Figure 1 microorganisms-11-01652-f001:**
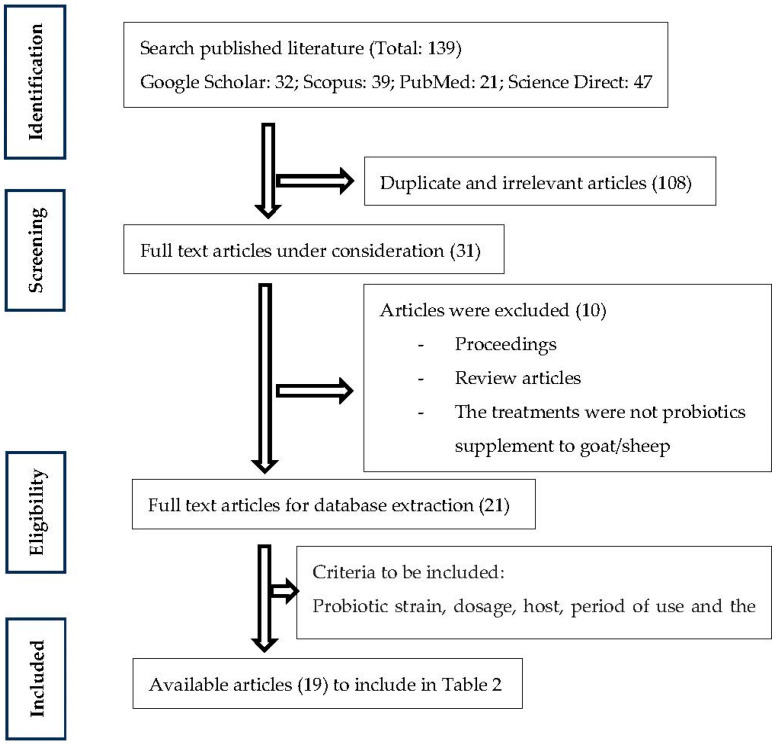
Flowchart of scientific literature search and selection for this review.

**Figure 2 microorganisms-11-01652-f002:**
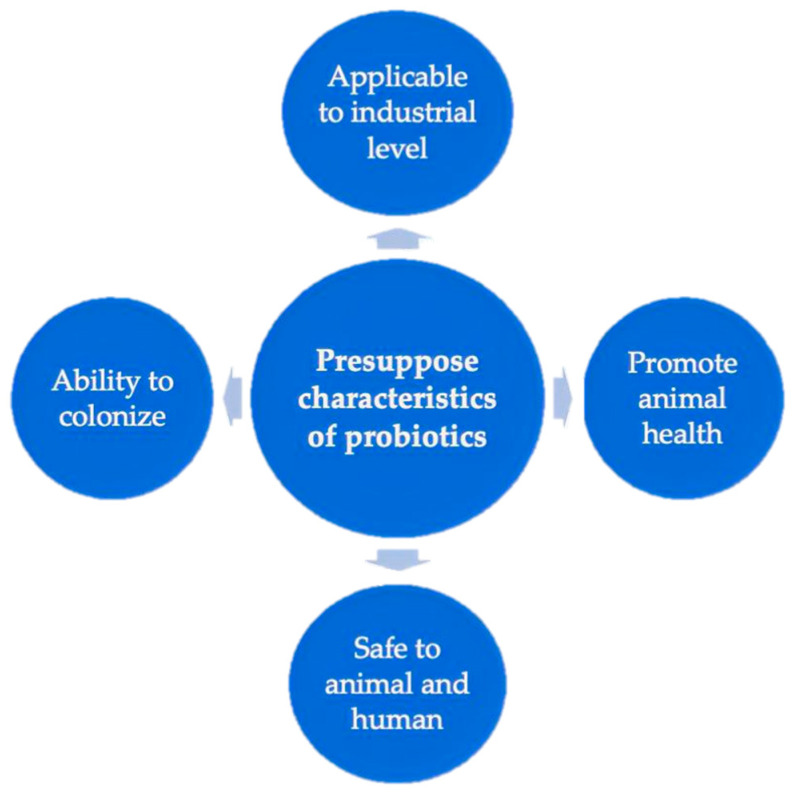
Considerable characteristics of probiotics.

**Figure 3 microorganisms-11-01652-f003:**
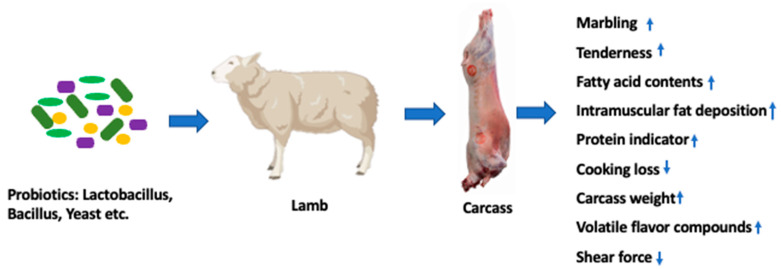
Effect of dietary supplement of probiotics on meat quality of lambs. (Created with BioRender.com). Abbreviations: ↑, increased; ↓, decreased.

**Table 1 microorganisms-11-01652-t001:** List of microorganisms commonly used as probiotics in small ruminants ^1^.

Genus	Species	Genus	Species
*Lactobacillus*	*L. acidophilus*	*Enterococcus*	*E. faecium*
	*L. plantarum*	*Pediococcus*	*P. acidilactici*
	*L. rhamnosus*	*Bacillus*	*B. subtilis*
	*L. reuteri*		*B. licheniformis*
	*L. casei*		*B. mesentericus*
	*L. brevis*		*B. cereus*
	*L. delbrueckii sub* sp. *bulgaricus*		*B. toyoi*
	*L. salivarus*	*Aspergillus*	*A. oryzae*
*Lactococcus*	*L. lactis*		*A. niger*
	*L. cremoris*	*Saccharomyces*	*S. cerevisiae*
*Bifidobacterium*	*B. bifidum*		*S. boulardi*
	*B. animalis*		
	*B. thermophilum*		
	*B. pseudolongum*		

^1^ Source: Abd El-Trwab et al. (2016) [28]; Seo et al. (2010) [30].

**Table 2 microorganisms-11-01652-t002:** Effect of probiotics on carcass and meat quality of small ruminants.

Probiotics	Administration /Dosage	Host	Duration	Effects on Carcass/Meat Quality	References
Se-yeast (*Saccharomyces cerevisiae*); Cr-yeast (*Saccharomyces cerevisiae*)	0.3 mg Se/d/head 0.25, 0.35 Cr/d/head	54 Rambouilet lamb	95 d	↑ meat quality ↑ retained protein in carcass ↑ total protein content in meat	[50]
Live yeast (*Saccharomyces cerevisiae*) (Yea Sac 1026)	5 g/d/head 1 × 10^8^ CFU/g	24 newly weaned Texel male lamb	62 d	↑ carcass weight ↑ carcass length ~ dressing percentage, compactness index, external chest depth, buttock circumference	[47]
Dairy yeast (*Kluyveromyces marximanus* NRRL3234; *Saccharomyces cerevisiae NCDC42*; *Saccharomyces cerevisiae ATCC9080*)	1 mL/kg body weight 1.5-2.0 × 10^9^ CFU/mL	60 lambs	91 d	~ carcass weight, carcass yield, carcass composition	[51]
*Lacticaseibacillus casei* HM-09 and of *Lactiplantibacillus plantaru* HM-10 (Probiotic supplement, Inner Mongolia Sci-Plus Biotech company, Beijing, China)	*L. casei* (1.5 × 10^9^ CFU/g) *L. plantarum* (1.5 × 10^9^ CFU/g)	24 Sunit lambs	90 d	↑total antioxidative capacity (T-AOC) and catalase (CAT) activity of LT muscle ↓ superoxide dismutase (SOD) activity ↑meat tenderness and flavor ↓shear force values and lightness in LT) muscle ~ pH, meat color (lightness, redness, yellowness) ~ cooking loss Alter the composition of meat volatile flavor compunds (nonanal, undecanal, 1-pentanol, 1-hexanol, and 2,3-octanedione) (observed by electric nose)	[12]
*Lactiplantibacillus plantarum* powder (Shandong Baolai Bioengineering Co., Ltd., Taian, China)	12 g/d/head 3 × 10^10^ CFU/g	12 Sunit sheep	90 d	↑ volatile flavor compounds (nonanal, decanal, and 1-hexanol) in tail fat ↑ Nonanal and decanal compounds in muscle	[13]
*Lacticaseibacillus casei* Zhang and *Lactiplantibacillus plantarum* P-8	1% level in diet 1.5 × 10^9^ CFU/g	12 Sunit lamb	90 d	↑ muscle production (myogenesis) and meat quality ↑ intramuscular fat deposition ↓ cooking loss and shear force ↑body length, LT area ↓ MYOD1 gene expression in muscle ↑ MAPK signaling pathway activity in muscle	[3]
*Saccharomyces cerevisiae* Se-Cr-yeast (*Saccharomyces cerevisiae*)	1.50 g/kg DM/d 1.65 × 10^10^ CFU/g	32 Pelibuey × Katahdin lambs	56 d	~ body weight ~ chop area, dorsal fat, carcass yield	[9]
Commercial probiotic feed “Amilotsin” *Lacobacillus* spp.	10 g/d/head	40 castrated Kalmyk breed rams	42 d	↑ carcass weight ↑ protein and energy value in meat ↑ unsaturated fatty acid composition in the fat of lambs ↑ smell, taste, juciness, and tenderness of meat (observed by panelists) ↑ organoleptic characteristics of meat ↑weight (liver, heart, lungs, kidney, and spleen)	[48]
Mixture of *Bacillus licheniformis, B. subtilis* and *Lactiplantibacillus plantarum* (ratio of 1:1:0.5)	10 g/kg feed	40 lambs (Chuanzhong blak)	70 d	↑ carcass yield ↑ abdominal fat% ↑ moisture and intramuscular fat in LT muscle ↑ mRNA expression of MyHC ↓ shear force value, lightness, yellowness, and heneicosanoic acid in LT muscle ↑ meat quality (color and tenderness)	[5]
Yeast; Lactic acid bacteria (*Lactobacillus* spp.)	2.5 g yeast/d/head 5 g yeast/d/head 2.5 g yeast + LAB/d/head 5 g yeast + LAB/d/head	18 West African Dwarf Buck	-	~ body weight, carcass weight ↓ cholesterol in carcass and liver ↑ quality of *Longissimus dorsi* muscle Affect color, juciness, tenderness, water-holding capacity, marbling of meat	[14]
*Saccharomyces cerevisiae*	0.5, 1.0 and 1.5 mL/kg body weight (3.6 × 10^9^ cells/mL)	16 Malpura lambs	180 d	~ carcass trait (weight and composition) ~ meat quality (cooking loss, chilling loss, and water-holding capacity)	[17]
*Saccharomyces cerevisiae* (*Yea Sace cepa 1026*) *Saccharomyces cerevisiae* + Selenium	3 g/d/head (5 × 10^6^ CFU/g)	24 male Texel lambs	120 d	↑ fat% in meat ~ moisture%, protein%, ash% ~ meat quality (color, cooking loss, shear force) ~ sensory traits (flavor, tenderness, and juciness) (observed by panelists)	[52]
*Lactobacillus reuteri* E81 *Lactobacillus rhamnosus* GG *Saccharomyces cerevisiae*	300/600 ppm of *Lactobacillus*/head (4 × 10^10^ CFU/g)	90 Anaolian Merino weanling lambs	70 d	↑ body weight, daily weight gain, and feed conversion ratio ↑ lightnessin meat ~ redness and yellowness in meat ↑ fattening performance of lamb ↑ meat pH value	[6]
*Saccharomyces cerevisiae* SC47	3 g/d/head 4.5 g/d head	27 male Zandi lambs	84 d	~ muscle fatty acid profile (saturated, unsaturated, monounsaturated, and polyunsaturated) ~ meat quality (% fat, protein, ash, pH, organic matter, and dry matter) ~ carcass traits (% muscle, bone, and fat)	[53]
Dry yeast (*Saccharomyces cerevisiae*); Soybean meal + dried yeast (*Saccharomyces cerevis*)	Dry yeast: 22.42% DM/d Soybean meal + dried yeast: 9.78% DM/d	27 goat kids (18: 3/4 Boer + ¼ Sanen; 9: Sanen)	154 d	~ carcass weight, weight loss by cooling, carcass yield, carcass compactness index	[54]
*Lactobacillus acidophilus* *Saccharomyces cerevisiae* (Probiotics, L.P. Feeds Tech Co., Ltd., Bangkok, Thailand)	2.5 and 5.0 g/h/d *L. acidophilus* 2.0 × 10^12^ CFU/g and *S. cerevisia* 5.0 × 10^11^ CFU/g	30 growing goats (Thai native × Anglo-Nubian)	56 d	↑ conjugate linoleic acid, total n-6 and total poly-unsaturated fatty acids in plasma ↑ rumen metabolism and growth performance	[49]
*Streptococcus faecalis T -110* *Bacillus mesentericus TO-A* *Clostridium butyricum TO -A* *Lactose* (*Probiotic mixture*)	5 g/d/kid *S.faecalis* T -110 (2 × 10^8^ CFU/g); *B. mesentericus* TO-A (2 × 10^6^ CFU/g) *C. butyricum* TO -A (2 × 10^6^ CFU/g)	30 male native goat kids	180 d	↑ pre-slaughter weight ↑ carcass weight ↑ dressed weight and dressing percentage ↑ head and stomach weight ↑ carcass yield	[11]
Dry yeast, *Lactobacillus acidophilus*, *Enterococcus faecium*, *Baciluus subtilis and Aspergillus oryzae*	*L. acidophilus* (2.5 × 10^9^ CFU/g) *E. faecium* (2.5 × 10^9^ CFU/g) *B. subtilis* (22,125 protein catalytic unit/g) *A. oryzae* (13,289 bacterial amylase unit)	63 Boer crossbred meat goats but 24 goats were used for carcass traits experiment	3 years (57 days for carcass traits experiment)	~ Carcass weights and weights of fabricated cuts (shoulder, loin, leg, rack, shank, and total parts) as well as carcass length, leg circumference, loin eye area, and backfat	[55]
*Bacillus subtilis* and *Bacillus licheniformis*	600 mg/kg 1 × 10^11^ CFU/g	39 male goats (Yantse River Delta White)	80 d	↑ pre-slaughter weight ↑ rib tissue thickness (GR)	[10]

Abbreviations: ↑, increased; ↓, decreased; ~, not changed, CFU, colony-forming unit; LT, Longissimus thoracis; MAPK, mitogen-activated protein kinase; MyHC, myosin heavy chain; DM, dry matter; LAB, lactic acid bacteria.

## Data Availability

Not applicable.

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
