# Peer review of "Recent Advances in the Use of Probiotics to Improve Meat Quality of Small Ruminants: A Review"

_microorganisms, 2023, doi:10.3390/microorganisms11071652_

Round 1

Reviewer 1 Report

Dear Authors,

Thank you for sending your manuscript. Attached please find comments and suggestions to improve the manuscript before it can be considered for publication. 

Topographical errors are seen throughout the manuscript. Authors should proofread the revised manuscript.

Author Response

Dear Reviewer,

We are grateful to you for your insightful comments on our paper. Please see the attached file for the responses to the reviewer.

Thank you.

Reviewer 2 Report

The manuscript entitled "Recent advances in the use of probiotics to improve meat quality of small ruminants: A review" provides valuable information and may be published in " Microorganisms". The article is very well prepared and written clearly.

In the article, I would only recommend adding the complete list of abbreviations and a graphical abstract.

Author Response

Response to Reviewer 2

The manuscript entitled “Recent advances in the use of probiotics to improve meat quality of small ruminants: A review” provides valuable information and may be published in “Microorganisms”. The article is very well prepared and written clearly.

AU: Thank you very much for the positive comments.

In the article, I would only recommend adding the complete list of abbreviations and a graphical abstract.

AU: We have followed the suggestions to improve the manuscript. We have added the abbreviation and a graphical abstract in the manuscript.

Reviewer 3 Report

The article provides general information on the effect of probiotics on the quality of the carcass and meat of small ruminants. Research on the quality of livestock meat consists of a number of characteristics such as: pH, meat color, texture (TPA), fatty acid profile, etc. The correct values of these traits varies depending on the species of animals from which they are collected. The manuscript includes information on the positive or negative effect of probiotics on the trait. However, information on the correct values in small ruminants is missing, and it is not stated by how much the use of the additive improved/deteriorated meat quality.

From the literature data collected, the authors presented the data in a table, where we can see a large variation in the amount of probiotic added and the length of animal rearing. 

The probiotic values are given in different units (mg, g, ml/kg, g/kg, mg/kg) which makes the interpretation of the results very difficult. 

The paper should be rebuilt, with detailed information on standard pH, meat color, TPA, etc. for small ruminants and the changes that occurred after probiotic application (how much difference?). This way the reader will know to what extent the value of a given trait will change. 

Author Response

Response to Reviewer 3

The article provides general information on the effect of probiotics on the quality of the carcass and meat of small ruminants. Research on the quality of livestock meat consists of a number of characteristics such as: pH, meat color, texture (TPA), fatty acid profile, etc. The correct values of these traits varies depending on the species of animals from which they are collected. The manuscript includes information on the positive or negative effect of probiotics on the trait. However, information on the correct values in small ruminants is missing, and it is not stated by how much the use of the additive improved/deteriorated meat quality.

AU: Thank you. We agree with you, meat quality consists of several characteristics, such as pH, meat color, texture (TPA), fatty acid profile, etc. we have included the effects of probiotics on these traits. In this manuscript, mainly we have focused on any effect of probiotics was seen on performance and meat quality when probiotics used as a dietary supplement to the small ruminants. All experimental studies included in the Table 2 provided the doses of probiotics used to observe their effect on meat quality.

We have a plan to write another review in future to separate the goat and sheep species, with including the details values (% of BW increase, % carcass yield increase etc…)

From the literature data collected, the authors presented the data in a table, where we can see a large variation in the amount of probiotic added and the length of animal rearing.

AU: Thank you. Every study has different experimental design, and used of probiotics species in their study, so it would be differed (vary) from one study to another study.

The probiotic values are given in different units (mg, g, ml/kg, g/kg, mg/kg) which makes the interpretation of the results very difficult.

AU: We mentioned the different probiotic values are given in different units (mg, g, ml/kg, g/kg, mg/kg) in different studies because the different original study used different probiotic species with different doses and even length of study was different. However, we can easily interpret (control vs probiotic group) the results within the study.

The paper should be rebuilt, with detailed information on standard pH, meat color, TPA etc. for small ruminants and the changes that occurred after probiotic application (how much difference?). This way the reader will know to what extent the value of a given trait will change.

AU: Thank you. We have included the details information of meat quality traits. Is probiotics have any effect on meat quality. We have a plan to write another review in future to separate the goat and sheep species, with including the details values (% of BW increase, % carcass yield increase, pH, meat color etc…)

Reviewer 4 Report

Microorganisms-2347615

General comments:

On the whole, the review of prior papers in the field is comprehensive without being overly long. There could be more discussion about the studies listed in Table 2 to form a hypothesis for future studies regarding meat quality.

English needs significant editing, preferably by a native speaker, as it is difficult to follow along

Interesting why Lambe et al. paper [ 55] was not included in Table 2. I understand the rest as they are not on sheep or goat meat.

Could break down the body of the review into sections, i.e., effect of probiotic supplementation on general eating quality traits L145-192, effects of probiotic supplementation on lipid oxidation L195-213

How relevant are the safety concerns? There is no mention of these in the conclusions

Specific comments:

L106 - change "El-Tawab" to "El-Trwab"

L195 - add "(2016)" after Ozer et al.

L208 - add "(2019)" after Trebelsi et al.

L209 - what species of meat were used in these sausages?

L284 - "2019" should be in bold

L295 - add page numbers 57-65

L313 - add volume number 11

L338 - add page numbers 893-906

L351 - change "656" to "8, 681389"

L363 - change "slam" to "Islam"

L438 - add volume and page numbers "179, 1-89"

Grammar is poor throughout manuscript with the exception of figures and tables, please pay attention to use of plurals, tense and verb conjugations. Would recommend a native speaker read over the manuscript prior to re-submission.

Author Response

Dear Reviewer,

We are grateful to the reviewer for your insightful comments on our paper. Please see the attached file for the response to reviewer.

Thank you!

Reviewer 5 Report

This is an interesting and relevant topic. At this stage, however, I recommend the authors to follow the guidelines for systematic reviews as per instructions for authors. The list of studies presented in Table 2, can only make sense if presented in the context of a systematic review.

I would recommend editing the English language by someone proficient. I am not a native but I have detected many editing needs. I am not detailing these as the list would be extensive. I would be happy to review in the second round.

Author Response

Dear Reviewer,

We are grateful to the reviewer for your insightful comments on our paper. Please see the attached file for response to reviewer.

Thank you!

Round 2

Reviewer 1 Report

Dear authors,

Thank you for submitting your revised manuscript. I have gone through the corrections you made, and I am happy with them. 

The English language has been improved on the revised manuscript. However, minor editing will be required to improve the readability of the paper.

Author Response

Dear Reviewer,

Thank you so much for your nice comments and consideration given to our revised version manuscript.

We have tried to improve the readability in our new revised version of the through minor English editing process. We hope, now it will meet your requirements.

Best regards

Authors

Reviewer 4 Report

Microorganisms-2347615

In the second revision this manuscript has been greatly improved for English. It provides a comprehensive overview and critically evaluates recent works in the field. Only minor grammatical errors are recommended prior to subsequent revisions.

I would consider changing keywords to not use words already within the title

Reference 37 (Schierack et al., 2009) is not cited within the manuscript. Fuller (1989) is not in the reference list.

L17 – add “is” before considered

L18 – add “from” before probiotics

L35 – add “is expected to” before rise

L61 – remove “meat especially red meat”

L72 – add “have been” after Probiotics

L74 – change “enteric diseases prevention” to “prevention of enteric diseases”

L80 – move “(Saccharomyces cerevisiae)” to after yeast

L96 – add “and” before small

L96 – how did you determine an optimal age range for articles? And why “before 2023”? In Table 2 (mentioned in L104) there are articles from 2007-2022

L229 – lower case p for probiotic

L232 – add reference for this statement (Fuller 1989, Probiotics of man and animals, Journal of Applied Bacteriology 66(5): 365-378)

L236 – change “FAO/WHO”to “the FAO and WHO” before [24]

L236 – change ”FAO/WHO” to “these organisations” before probiotics

L238 – change “Food and Drug Administration, USA” to “the USA Food and Drug Administration”

L240 – add “be” before metabolically

L241 – add “and” before safety

L246 – change “The genus Lactobacillus and Bifidobacterium reported as bacterial” to “The Lactobacillus and Bifidobacterium genera are reported as the bacterial”

L249 – change “Bacillus is” to “Bacillus sp. are”

L251 – remove “Within”

L255 – change “growth promoter, promote animal health” to “growth promotion, animal health promotion”

L285 – add space between “their” and “health”

L286 – remove “the” before animals

L289 – remove ”as” bfore complementary

L291 – add “and” before specific, change “influencing” to “influence”

L292 – remove colon after microorganisms

L295 – change “maintain” to “maintaining”

L298 – change “enhanced” to “enhance”

L303 – change “provides” to “provide”

L318 – change “regulate” to “regulating”

L319 - change “regulate” to “regulating”

L448 – add “Instead,” before most

L453 – remove “factors such as”

L507 – add “and” before indicate

L508 – change “decision” to “decisions”

L509 – change “are” to “depend on”

L511 – change “presence of pH” to “pH level”

L512 – change “can a decrease in color intensity” to “can decrease the color intensity”

L515 – add “sp.” after Lactobacillus

L516 – add “and” before yeast

L517 - add "and” between odor and flavor, change comma to semicolon after meat

L519 – change “causes to” to “and thereby”

L548 – change “stage” to “stages”

L550 – change “carcass” to “carcasses”

L553 – consider changing this sentence “Oxidation of myoglobin might reduce the color of meat” to “Oxidation of myoglobin reduces the redness of meat”

L558 – change “catch’ to “holding”

L559 – add a reference for “The water holding capacity of meat..” is it Tian et al. as well?

L560 – change “Diet” to “Diets”

L562 – add “been” before positively

L563 – change “supplement” to “supplementation”

L564 – change “deposited” to “deposit”

L565 – change “to soft” to “softness”

L566 – add “other” before studies

L567 – combine these sentences using a semicolon

L568 – remove “meat”, change “probiotics” to “probiotic”

L573 – add “It can be concluded that” before “the use of probiotics..”

L633 – change “Relate” to “Related”

L639 – change “higher” to “greater”

L641 – change “lambs” to animal”

L642 - change “doses” to “dosage” and “when we” to “which are”

L642 - change “lambs” to animal” and start sentence with “In most of the cases probiotics are considered safe..”

L645 – change “relate with” to “related to”

L646 – change “probiotics are very barely” to “probiotic genera are barely”

L651 – what is meant by “one probiotic or specific probiotic”?

L657 – add “to” between this and the

L660 – remove ”Because” and change “associate” to “occur”

L814 – change “assessment” to “assessments”, “microorganisms” to “microorganism” and “use” to “used”

L827 – change “meat quality in small ruminants” to “quality of meat from small ruminants”

L855 – add”-527” after 523

L891 – change “1-12” to “112”

L964 – remove second instance of “-e1539119563”

L1006 – change comma to full stop after Animal

L1008 – change comma to full stop after “Naringsforskning”

Author Response

Dear Reviewer,

Thank you for your precious time in reviewing our paper and providing valuable comments. Please see the attached file for the replies of your comments. 

Best regards

Authors

Reviewer 5 Report

The authors did not address the issue raised. Please note, as per instructions for authors you need to follow the PRISMA guidelines. Links for guidance are therein provided. I will not process further this review until these changes are made.

>

Author Response

Dear Reviewer,

Thank you. We have changed according to your suggestions. Anyway, in this review we followed the PRISMA guidelines for selecting the articles, however, the meta-analysis was not performed. The literature was selected based on the following criteria mentioned in Figure 1 in our revised manuscript (Line 90-114)

Best regards

Authors